# A Preliminary Study: Evaluation of Oral Trazodone as a Strategy to Reduce Anesthetic Requirements in Bitches Undergoing Ovariectomy

**DOI:** 10.3390/ani15060854

**Published:** 2025-03-17

**Authors:** Nerea Cambeiro-Camarero, Silvia Fernández-Martín, Antonio González-Cantalapiedra

**Affiliations:** 1Department of Anatomy Animal Production and Veterinary Clinical Sciences, Faculty of Veterinary, Universidade de Santiago de Compostela, 27002 Lugo, Spain; antonio.cantalapiedra@usc.es; 2Rof-Codina Veterinary Teaching Hospital, Faculty of Veterinary, Universidade de Santiago de Compostela, 27002 Lugo, Spain

**Keywords:** trazodone, stress, pain, ovariectomy, analgesia, anesthesia

## Abstract

Trazodone is an anxiolytic drug used in veterinary medicine to minimize stress in patients during visits to the clinic. This study evaluated the positive effect of stress reduction through trazodone administration as part of premedication in routine surgical procedures. It was assessed whether it could reduce the impact of intraoperative painful stimuli, whether it allowed for a reduction in the dose of induction drugs, and whether its use in the surgical environment was safe and free of adverse effects. Two groups of animals participated in this study, eight receiving trazodone and the other eight a placebo. It was observed that the animals that took trazodone were less anxious, and the required doses of induction agents were lower than in the control group. Further studies are required to better assess the implications of using trazodone in anesthetic protocols, with pre-surgical stress reduction being an aspect that should receive more consideration due to its possible beneficial effects in veterinary medicine.

## 1. Introduction

In veterinary medicine, a balanced anesthesia is achieved by combining various drugs to enhance desired effects while minimizing adverse effects, which increase directly with higher drug doses [1]. A good balanced anesthesia provides an adequate depth of anesthesia combined with analgesia that allows the patient to undergo unpleasant procedures while maintaining proper cardiovascular and respiratory function [2]. To make this possible, pre-surgical patient management is crucial even before the patient arrives at the hospital [3].

The stress, fear, and anxiety many patients experience upon arriving at a veterinary hospital must be considered, as they are factors that can increase pain perception [3]. Additionally, animals that experience fear, stress, and anxiety are more likely to display behaviors that may result in harm to the veterinary staff [4,5]. Therefore, less stressed animals are less likely to cause harm and are more likely to undergo a more thorough physical exam, which ensures the patient’s condition is assessed before surgery [1].

To reduce stress and associated behaviors, drugs such as trazodone can be used. Trazodone acts on the serotoninergic system as an antagonist of 5-HT2A, 5-HT2C, and 5-HT7 receptors and as a partial agonist of the 5-HT1A receptor. It also moderately inhibits serotonin reuptake and blocks α1-adrenergic and histamine receptors [6]. Its action on various serotoninergic receptors enables trazodone to produce anxiolysis, regulate the stress response, and improve the mood of the treated patient [7]. In veterinary medicine, trazodone is already used as a sustained treatment for managing behavioral disorders caused by fear and anxiety [8]. It is also used as a single-dose treatment before veterinary visits to facilitate handling and minimize stress-related physiological parameter alterations during veterinary visits [9]. In hospitals, it is used to reduce stress in hospitalized animals [10].

The most observed adverse effects of this drug include dizziness, headache, drowsiness, and dry mouth [11]. In human medicine, other side effects include the possibility for arrhythmias in patients with pre-existing heart conditions, as trazodone can delay cardiac repolarization in a dose-dependent manner. It can also cause hypotension due to α1-adrenergic receptor blockade, especially in patients concurrently receiving antihypertensive therapy, with underlying heart disease, or in geriatric patients [6].

This study aims to evaluate the potential benefits of trazodone as part of premedication for ovariectomy via median laparotomy. Specifically, it assesses whether trazodone can reduce stress during preoperative management, decrease the propofol requirements for induction, and minimize the need for fentanyl IV boluses while monitoring HR and NIAP throughout the surgical procedure.

## 2. Materials and Methods

A prospective clinical, randomized, double-blinded study was conducted on 16 adult female dogs that visited the Rof-Codina Veterinary Teaching Hospital (RCVTH) (Lugo, Spain) for scheduled ovariectomy surgery between November 2023 and November 2024. All procedures were approved by the Ethics Committee of the Galician Public Foundation Rof Codina (Reference number: AELU001/23/FUN (01)/OUTROS (11)/AGC/01).

To ensure the homogeneity of groups, the parameters age, temperament, weight and body condition score (BCS) were assessed. For the BCS assessment, a scale of one to five was used, with one being thin and five being obese [12]. During the physical examination, heart rate (HR; beats/min), respiratory rate (RR; breaths/min), non-invasive arterial pressures (NIAP; systolic; SAP; mmHg, diastolic; DAP; mmHg, mean; MAP; mmHg), femoral pulse (FP; strong, regular, synchronous/weak, irregular, asynchronous), capillary refill time (CRT; seconds), mucous membrane color (pale/rosy/red/blue, wet/dry), and temperature (T; °C) were evaluated. The presence of vomiting after premedication was also recorded.

All animals underwent an analytical study via venipuncture of cephalic vein (3 mL per sample) before surgery and after premedication. The study included a complete blood count and a routine preanesthetic biochemical study.

The animals were classified according to the anesthetic risk of the American Society of Anesthesiologists (ASA), with those classified as ASA > I being excluded. Also, animals showing any surgical or anesthetic complications or requiring specific medical or dietary treatment were also excluded. All animal guardians signed written informed consent, which included a detailed explanation of the study protocol. The surgical procedures and anesthetic monitoring were performed by the same highly qualified veterinary surgical and anesthesia team. In some cases, staff from the Rof-Codina Veterinary Teaching Hospital (RCVTH) and veterinary students collaborated in preparing the animals for surgery and postoperative care. The animals were randomly assigned to two groups based onthe outpatient medication received: the Trazodone Group (TG) (*n* = 8) received 5 mg/kg of trazodone (Trazodone Normon, Normon Laboratories S.A.) orally (PO), and the Control Group (CG) (*n* = 8) received an inert placebo orally (PO), both groups received either trazodone or placebo approximately 2 h before the animals had arrived at RCVTH. All animal guardians received the pill the day before surgery, and when the animals arrived at the hospital, the pre-surgical procedure immediately started for all patients.

### 2.1. Anesthesia and Surgery Protocol

The animals were premedicated in a calm environment with a mixture of dexmedetomidine (5 µg/kg IM, Dexdomitor, Ecuphar) and morphine (0.3 mg/kg IM, Morphine, BBraun). Once the animal achieved light-to-moderate sedation, a cephalic vein catheter was placed, followed by blood collection and the administration of an intravenous (IV) Lactated Ringer’s solution (Lactato Ringer Vet, BBraun) at an infusion rate of 5 mL/kg/h. Before surgery, the animals received an antibiotic, cefazolin (22 mg/kg IV, Cefazolin Normon, Normon Laboratories S.A.), which was administered at the same dose 90 min later. After cleaning and shaving the surgical field, the animals were moved to the operating room, where oxygen was administered via a mask at 3–4 L for a few minutes. Induction was performed with propofol (0.5–3 mg/kg IV, Lipuro, BBraun, Melsungen, Germany) followed by orotracheal intubation. The animals were placed in dorsal recumbency, and controlled volume mechanical ventilation was started using an anesthesia machine (Mindray, WATO EX-35). Tidal volume was initially set at 10 mL/kg, then adjusted to maintain normocapnia (EtCO_2_ = 35–45 mmHg) with a respiratory rate of 12–14 breaths per minute, an inspiration-to-expiration ratio of 1:2, a PEEP of 4 cm H_2_O, and an inspiratory pause at 25%. Anesthesia was maintained with sevoflurane (vaporizer setting at 2%) in 100% O_2_, with the concentration adjusted to achieve the appropriate anesthetic depth, which was determined by clinical indicators, including mandibular muscle tone, ventromedial rotation of the eyeball, and the absence of the palpebral reflex [13]. Single dose of intraoperative meloxicam (0.2 mg/kg IV, Metacam, Boehringer Ingelheim) was also administered.

A cranial midline infraumbilical laparotomy was performed through a 4–8 cm incision (depending on the patient’s size). Each ovary was exteriorized applying gentle traction and resection was performed using a harmonic scalpel. After verifying the absence of hemorrhage, the incision was closed in three layers, muscular, subcutaneous and skin.

### 2.2. Behavioral Parameters Observed

To assess the level of stress, a reproduction from the “The Clinic Dog Stress Scale” (CDSS) [4,14] was used to objectively evaluate the animal’s behavior during initial interaction and physical examination. The scale (Table 1) included observations of body posture, ear position, gaze, breathing pattern, lip position, motor activity, and vocalizations. The maximum score of 27 points indicated the highest stress level (range 0 to 27).

A Visual Analog Scale (VAS) was also applied [15], where the clinician subjectively rated stress on a scale from 0 to 10, with 10 indicating very high stress.

Lastly, a shortened version of the reactivity evaluation form (REF) [16,17] was used to assign a score of one to four to dogs based on their behavior when the observer was directly in front of the animal, with the highest score indicating the highest level of avoidance and refusing every type of contact and the lowest score indicating stops immediately in front of the clinician and seeks contact.

### 2.3. Pre-Induction Sedation Assessment

The degree of pre-induction sedation was assessed using a reproduction of the Sedation Scale (SS) [18,19], evaluating seven items: spontaneous posture, palpebral reflex, eye position, relaxation of the jaw and tongue, response to noise (clap), resistance to lateral recumbency, and general appearance/attitude. The maximum possible score was 20 points (range 0 to 20), with higher scores indicating greater sedation (Table 2).

### 2.4. Induction Agent Requirements

In both groups, the propofol requirements for anesthetic induction were evaluated, with doses from 0.5 mg/kg to 3 mg/kg, to determine the minimal effective dose for orotracheal intubation. Propofol was administered in incremental boluses of 0.5 mg/kg over 15 s until orotracheal intubation could be achieved. After each bolus of propofol, intubation conditions were assessed. The anesthesia team recorded the total required dose to achieve specific endpoints, including loss of the laryngeal reflex, loss of palpebral reflex, decreased heart rate and respiratory rate, jaw relaxation and ventro-medial eye position with slight mydriasis.

### 2.5. Intraoperative Variables Observation

Respiratory and hemodynamic parameters were recorded at six perioperative time points: pre-induction (P), anesthesia onset (T0), abdominal cavity opening (T1), first ovarian pedicle traction (T2), second ovarian pedicle traction (T3), abdominal cavity closure (T4), and recovery (D), using a multi-parameter monitor (Mindray iPM12 Vet). The anesthesia team manually recorded the parameters every 5 min. The parameters recorded included the following: EtCO_2_ (mmHg), end-tidal sevoflurane concentration (FE’Sevo; %), respiratory rate (RR; breaths/min), HR; beats/min, NIAP: (SAP; mmHg, DAP; mmHg, and MAP; mmHg), oxygen saturation (SpO_2_; %), and T; °C.

### 2.6. Intraoperative Rescue Analgesia Requirements

Additionally, the anesthesia team monitored for signs of intraoperative pain every 5 min, determined by the appearance of spontaneous breathing and increases of 20% in SAP, MAP, DAP, and/or HR compared to baseline values at the beginning of anesthesia, stabilized for more than 30 s [20]. In such cases, a bolus of fentanyl (2.5 µg/kg IV, Fentadon, Dechra) was administered.

### 2.7. Statistical Data Analysis

The results were expressed as mean ± standard deviation (SD), and statistical analyses were performed using Sigma Plot 12.5 (Systat Software Inc., Chicago, IL, USA). Data normality was assessed using the Shapiro–Wilk test, and Levene’s test was used to evaluate the equality of variances for normal variables.

A comparison of variables between the two study groups (Trazodone Group and Placebo Group) was performed using the Student’s *t*-test for normal distributions and the Mann–Whitney U test for non-normal distributions. Cardiovascular and respiratory values were analyzed using one-way repeated-measures ANOVA, with a post hoc Holm-Sidak test. For non-normal variables, statistical comparisons were conducted using the Friedman repeated-measures analysis followed by a Tukey post hoc test. A *p*-value of <0.05 was considered statistically significant.

## 3. Results

Procedures were performed at the facilities of the RCVTH. Due to the inability to conduct a full physical examination, one of the selected animals failed to meet the inclusion criteria. To maintain the target number of female dogs in the CG, it was replaced with another female dog.

### 3.1. Animals

No significant differences were observed in terms of age, body weight and BCS (Table 3).

### 3.2. Physiological Parameters

During the physical examination, one of the eight dogs in the CG (12,5%) did not allow a complete physical examination, as she was excessively nervous. However, all dogs in the TG allowed the examination without issues.

The mean HR of the dogs in the TG was 125.75 ± 14.40 beats/min, and the mean HR of the dogs in the CG was 124 ± 25.78 beats/min, resulting in a *p* = 0.87, indicating no differences between the groups. The mean T of the TG was 38.63 ± 0.47 °C, and the mean temperature of the CG was 38.40 ± 0.62 °C, resulting in a *p* = 0.43, with no significant difference between the groups.

### 3.3. Hematological and Biochemical Parameters

Regarding the biochemical results, LAC, CREA, BUN, PT, ALB, WBC, and liver enzymes, no significant differences were observed between groups, and the values were within normal ranges. The results for HTC, HGB, and reticulocytes were within normal ranges for both groups, with no significant differences. The same applied to platelets and the various white blood cell types (Table 4).

### 3.4. Behavioral Parameters

The animals in the TG had a mean score of 5.12 ± 2.95 points out of 27 in CDSS, while the animals in the CG had a score of 11.50 ± 6.16 points out of 27. This represents a statistically significant difference between the two groups (*p* = 0.02). The VAS results showed that the dogs in the TG had a mean score of 3.87 ± 1.88 out of 10, and the dogs in the CG had a mean score of 6.00 ± 2.67 out of 10. This did not reach statistical significance (*p* = 0.09), but the TG results were lower than those of the CG. The REF results showed scores of 1.37 ± 0.52 out of 4 for the TG and 2.25 ± 1.16 out of 4 for the CG, with a *p* = 0.13, indicating no significant differences between the groups (Figure 1).

### 3.5. Premedication to Venoclysis Interval

The time between the administration of premedication and the sedation level required for successful venous catheterization for subsequent drug administration and fluid therapy was 11.62 ± 4.93 min for the TG and 14.75 ± 6.76 min for the CG. No significant differences were found between the groups (*p* = 0.23). It was observed that 62.5% of the animals in the CG vomited after IM premedication with morphine, whereas only 37.5% (3 out of 8) of the animals in the TG vomited after injection.

### 3.6. Induction Agent Requirements and Induction Data

The dogs in the TG had a mean score of 10.62 ± 5.04 out of 20 in the SS, while the dogs in the CG had a mean score of 8.12 ± 4.19, with no significant differences (*p* = 0.30). The time between premedication and induction for the TG and CG was 44.37 ± 8.21 min and 42.50 ± 8.86 min, respectively, with no significant differences (*p* = 0.67). The doses of IV propofol were 1.36 ± 0.55 mg/kg for the TG and 2.13 ± 1.20 mg/kg for the CG. This did not reach statistical significance (*p* = 0.28), but the TG results were lower than those of the CG. In both groups, 25% of the animals had difficult intubations after more than two attempts.

### 3.7. Intraoperative Rescue Analgesia

During surgery, the rescue analgesia requirements were 0.87 ± 0.64 boluses on average for the TG and 1.37 ± 0.92 boluses for the CG (*p* = 0.23), with no significant differences between the groups. However, the rescue analgesia boluses (fentanyl 2.5 µg/kgIV) were slightly higher in the CG.

### 3.8. Intraoperative Vital Signs Monitoring

No significant differences were observed between the HR at the same time points between the groups. Both groups showed a significant increase in HR at T1, referring to the opening of the abdominal cavity, compared to time P. However, during T2, the TG showed a lower HR (83.87 ± 14.02 beats/min) compared to the CG (94.62 ± 13.14 beats/min). The same was observed at T3, with an HR of 75.62 ± 20.94 beats/min for the TG and 84.00 ± 21.67 beats/min for the CG (Figure 2).

No significant differences were observed between the groups for NIAP. The increase in SAP at T2 in the CG (113.25 ± 16.55 mmHg) compared to the TG (106.57 ± 10.15 mmHg) was not statistically significant, however higher values were observed for the CG. For MAP at T2, the values were 76.57 ± 12.62 for the TG and 88.50 ± 12.17 for the CG (Figure 3). All recorded cardiorespiratory parameters, including EtCO2, T and FE’Sevo, are shown in Appendix A.

### 3.9. Recovery

Upon waking from anesthesia, three animals from the TG (37.5%) and six animals from the CG (75%) required a rescue dose of dexmedetomidine (2 µg/kg IV).

## 4. Discussion

Numerous studies have assessed how the use of trazodone may positively influence veterinary clinic visits [11,21], as well as its usefulness in treating behavior disorders related to fear and anxiety in dogs [22]. However, to the authors’ knowledge, this is the first study to evaluate the effect of trazodone on pre-surgical stress, as well as its impact on propofol requirements, intraoperative fentanyl boluses, and non-invasive blood pressure.

In the first instance, regarding the choice of drug, one interesting and practical aspect of trazodone is its oral administration, which does not cause stress for the animal and can be given with food. It is also easy for the owner to administer. Additionally, it is a widely used drug, and its use is considered safe [23] for treating behaviors associated with fear, stress, and anxiety [8].

In relation to the demographic data, to ensure the homogeneity of groups, age was considered, ensuring that all the female dogs had gone through at least one estrous cycle before undergoing surgery [24]. It has been observed that OV or ovariohysterectomies (OHTs) performed before 6 months, particularly in large breeds (>20 kg), may more commonly promote the development of joint problems, urinary incontinence, recurrent genital tract infections, and the appearance of neoplasms [25]. Additionally, animals with similar BCSs were selected [26]. Since all of them were healthy animals, the same drug doses would be expected to produce the same response. Thus, obesity was avoided as a variable since it increases the anesthetic risk [27].

In relation to the behavioral parameters, a study in which trazodone was administered at doses of 9 to 12 mg/kg 90 min before a veterinary visit concluded that it helped minimize stress-associated behaviors in dogs, as well as physiological stress parameters. They found trazodone to be safe [9] and that it did not produce serotonin syndrome [28]. As we expected, in the present study, the CDSS showed a statistically significant difference between groups. Moreover, the evaluation of VAS and REF results in our study showed lower stress scores for the TG than for the CG. To further compare our results, another study showed a reduction in stress levels without altering the neurological examination. In the mentioned study, 32 animals were treated with doses of trazodone ranging from 6.25 to 8.60 mg/kg [21]. Moreover, in a study involving a hospital setting, the evaluation of a group of 120 animals showed that trazodone (4 to 12 mg/kg) significantly reduced stress [10]. It is notable that other studies tend to use higher doses, while in our study, a cautious dose of 5 mg/kg was used.

As a comment about the results regarding the premedication-venoclysis interval, the authors note that a difference was observed between the two groups in the occurrence of vomiting as an adverse effect of premedication with morphine: fewer vomiting incidents occurred in the TG. All animal owners confirmed they adhered to the fasting protocol suggested by the veterinarian, so fasting time was no longer considered a parameter for this statistical variable. Trazodone is a serotonin antagonist [29], while other drugs such as ondansetron are also serotonin antagonists, known for their antiemetic application through the antagonism of 5-HT3 receptors in the chemoreceptor trigger zone in the area postrema of the fourth ventricle of the central nervous system. This area is stimulated by most anesthetic, chemotherapeutic, and opioid drugs capable of inducing emesis [30]. Therefore, it remains to be determined whether trazodone might have an antiemetic effect similar to other serotonin antagonists.

In relation to the propofol requirements, a study involving a combination of alfentanil and atropine as premedication showed that adding trazodone to the mentioned premedication reduced propofol requirements [31]. Additionally, another study demonstrated that preanesthetic protocols minimizing stress in cats achieved optimal sedation in a shorter time and required lower propofol doses as an induction agent compared to cats not receiving low stress protocols [32]. As expected, in our study, a reduction of the required dose of propofol in the TG compared with the CG was observed. Although this difference was not statistically significant, a trend in the results was observed.

Regarding the requirements of intraoperative rescue analgesia boluses, it is relevant to mention that it is not yet considered in veterinary clinics that all the physiopathological alterations caused by stress could affect the surgical patient’s pain response [3]. However, in human medicine, it is well known that stress plays a crucial role as a primary cause of pain and must be considered in clinical practice [33]. Studies have shown that patients with elevated stress and anxiety levels, measured using the Amsterdam Preoperative Anxiety and Information Scale (APAIS) [34], require higher doses of analgesics and experience increased postoperative pain [35]. Therefore, various techniques are used to minimize stress, such as music, which has been shown to reduce the neuroendocrine stress response in surgical patients [36].

A study about how stress can be an etiological factor in pain in children demonstrated that stress affects multiple brain regions. It is noted that stress increases muscle tone and promotes neuromuscular excitability. This is suspected to contribute to pain as another neurobiological mechanism in response to stress, making pain a signal of homeostatic imbalance that increases proportionally to stress [33]. In our study, it is noteworthy that, in two cases (one from each group, both with the lowest stress levels), no rescue analgesia was required during surgery. Those bitches also woke up without disorientation or pain and thus did not require additional premedication boluses upon awakening. It is not possible to draw conclusions from these data, but the observation remains open on how stress might affect the management of pain in surgical veterinary patients.

Regarding the NIAP values registered in our study, no significant hypotension SAP < 80–90 mmHg, DAP < 40 mmHg, or MAP < 60–70 mmHg [1] was observed. Similarly, non-hypotension directly related with the administration of oral trazodone (5 mg/kg for patients over 10 kg and 7.5 mg/kg in patients under 10 kg) was recorded in a study of 30 dogs undergoing orthopedic surgical procedures [37].

As for other side effects of trazodone, a study showed that trazodone (5 to 7.5 mg/kg) administered every 12 h reduces platelet aggregation, but no alterations were found in oral mucosal bleeding time, platelet counts, or coagulation times, so it remains inconclusive whether this could have clinical implications [38]. In our study, the role of trazodone in hemostasis was not quantitatively assessed; however, no abnormal intraoperative bleeding was observed.

The limitations of this study are as follows: since trazodone was administered at the animal’s home, this made it impossible to guarantee proper administration. Additionally, the study aimed to work with the most cautious doses possible, which likely resulted in an incomplete visualization of the real outcomes at higher, more effective doses. In addition, the pharmacokinetics and pharmacodynamics of trazodone were also not estimated, so the administration times were based on previous studies [39]. Moreover, blood samples were not collected after the surgery to evaluate hematological or biochemical parameters that could have shown differences between groups.

Therefore, this line of research remains open, with questions yet to be answered regarding the use and administration of trazodone or other drugs for managing stress in surgical patients and their implications for safety, health, and well-being throughout the entire surgical process.

## 5. Conclusions

The administration of oral trazodone two hours before arrival at the veterinary hospital for scheduled ovariectomy reduces stress during preoperative management. Additionally, animals receiving trazodone arrived at the induction stage more sedated and required lower doses of propofol. Finally, its use was safe for the animals included in the study. However, further studies with larger sample sizes are needed to confirm these results and fully evaluate the role of trazodone in preoperative protocols.

## Figures and Tables

**Figure 1 animals-15-00854-f001:**
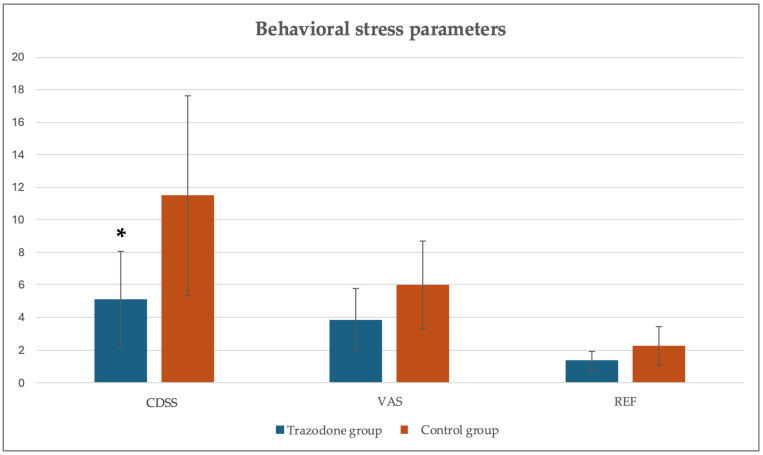
Behavioral stress parameters evaluated using the Clinical Dog Scale Stress (CDSS), a Visual Analog Scale (VAS) and a Reactivity Evaluation Form (REF). Values given as a mean ± SD. *p* < 0.05: * vs. control group.

**Figure 2 animals-15-00854-f002:**
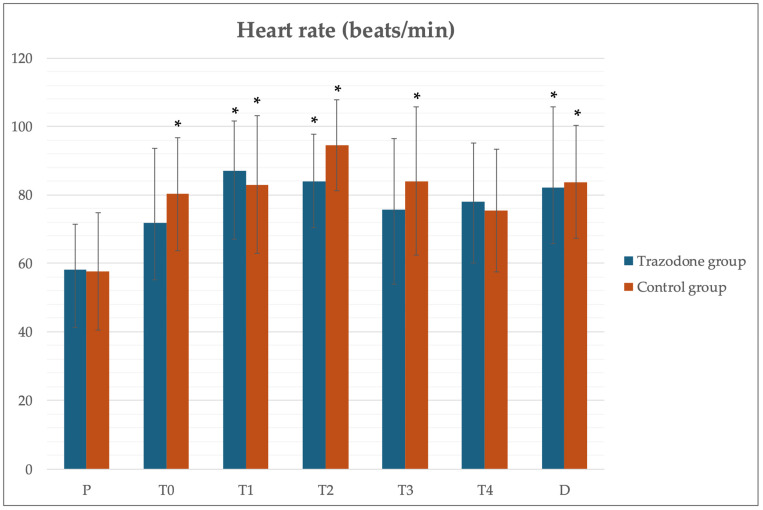
Heart rate values at different points in time during the experimental period. Values given as a mean ± SD. *p* < 0.05: * vs. pre-induction (P) within the same group.

**Figure 3 animals-15-00854-f003:**
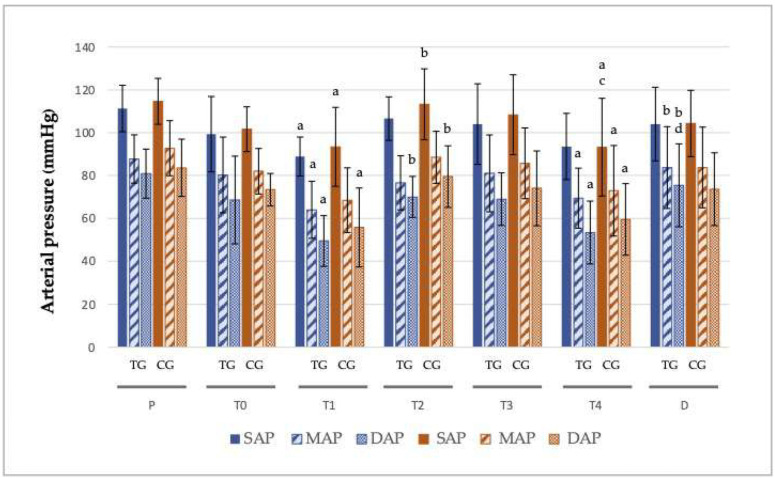
Non-invasive arterial pressure (NIAP: systolic; SAP; mmHg, diastolic; DAP; mmHg, mean; MAP; mmHg) at different points in time along the experimental period in the trazodone (TG) and control group (CG). Values given ± SD. *p* < 0.05: a vs. P, b vs. T1, c vs. T2, d vs. T4, within the same group.

**Table 1 animals-15-00854-t001:** The Clinic Dog Stress Scale, used to measure the dog behavior in the preoperative preparation room.

Stress Level	Body Posture	Ears Position	Gaze	Respirations	Mouth	Activity	Vocalization
0	Relaxed and moves on itself.	Upright and softly forward.	Look constantly at the vet.	Normal. Jaw relaxed.	Relaxed.	Moves naturally.	No.
1	Tense. Can be manipulated	Turned a little back.	Look intermittently at the vet	Normal. Jaw tensed.	Firm.	Inactive.	Crying.
2	Tense. Difficult to manipulate	Fully turned backward.	Does not look at the vet but stalks the place.	Panting without salivating.	Linking lips.	Flexed paws. Can tremble.	Whimpering.
3	Bent over and crouching. Cannot manipulate the ventral part.	Turned backward and down.	Look steadily at the distance.	Panting and dripping.	Masking and linking.	Periodic trembling.	Growling.
4	Completely crouched. Cannot be manipulated at all.	As backward and down as possible.	Look steadily at immediate fore-distance.	Hardly panting and salivating.		Trembling without stop.	Biting.

Clinic Dog Stress Scale (CDSS) reproduced from King et al. [4].

**Table 2 animals-15-00854-t002:** Sedation Scale.

Parameters	Score			
	0	1	2	3
Spontaneous posture	Stand up.	Tired but stand up.	Lying but can stand up.	Cannot stand up.
Palpebral reflex	Fast.	Slow but close the eye completely.	Slow. Cannot close the eye completely.	None.
Eyes position	Central.	Rotated down. None, hidden.	Rotated down. Hidden by third eyelid.	—
Tone of jaw and tongue	Normal. Strong nausea reflex.	Reduced tone. Moderate nausea reflex.	Very reduced tone. Barely appear nausea reflex.	Totally relaxed jaw. No nausea reflex.
Noise response	Turns head towards the noise. Shrinks.	Moderate turn of head. Minimum shrink.	Minimum response to noise.	Nonresponsive.
Resistance to lying in lateral.	Does not allow.	Fights but finally allow the position.	Barely fights. Allow the position.	No resistance.
Appearance/general attitude	Excitable.	Awake.	Calmed.	Stupor.

Sedation Scale reproduced from Wagner et al. [18], originally described by Grint et al. (2009) [19].

**Table 3 animals-15-00854-t003:** Mean ± SD of demographic data of the animals.

Parameter	Trazodone Group	Control Group
Age (months)	14.12 ± 8.36 (7–34)	12.62 ± 4.18 (8–20)
Weight (kg)	16.77 ± 8.14 (10.3–31)	13.65 ± 5.67 (3.9–22)
BCS [1–5]	2.37 ± 0.52 [2–3]	2.75 ± 0.71 [2–4]

**Table 4 animals-15-00854-t004:** Preoperative biochemical and hematologic analysis.

Variables	Trazodone Group	Control Group
GLU (mg/dL)	119.37 ± 16.77	107.25 ± 13.17
LAC (mg/dL)	1.87 ± 0.71	2.30 ± 0.48
CREA (mg/dL)	0.77 ± 0.13	0.87 ± 0.15
TP (g/dL)	6.01 ± 0.32	6.01 ± 0.36
ALB(g/dL)	3.15 ± 0.21	3.09 ± 0.20
GLOB(g/dL)	2.86 ± 0.30	2.92 ± 0.22
HTC (%)	44.17 ± 4.65	43.86 ± 4.22
HGB (g/dL)	16.21 ± 1.66	16.55 ± 2.77
RETIC (K/µL)	30.70 ± 16.14	29.76 ± 16.61
Leucocitos (K/µL)	12.76 ± 2.86	12.96 ± 3.11
PLQ (K/µL)	281.00 ± 35.65	248.25 ± 81.88

Venous blood data. Values given as mean ± SD. Statistically significant differences were considered when *p* < 0.05.

## Data Availability

The original contributions presented in this study are included in the article/Appendix A. Further inquiries can be directed to the corresponding authors.

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
