# Peer review of "A Preliminary Study: Evaluation of Oral Trazodone as a Strategy to Reduce Anesthetic Requirements in Bitches Undergoing Ovariectomy"

_animals, 2025, doi:10.3390/ani15060854_

Round 1
Reviewer 1 Report
Comments and Suggestions for Authors
The authors present an interesting report on the effects of trazodone administration prior to veterinary visits that involve general anesthesia. The patient numbers are quite low, and there does not appear to be a power analysis or justification for why 16 patients total were thought to be enough. There also seems to be quite a bit of interpretation outside of the actual statistical analysis - if 2 groups are not different based on your pre-determined analysis strategy, then they are not different... There is no such thing as a trend when rigorously evaluating 2 groups for differences - they are either different or they are not. The authors need to revise their language to reflect what the data shows, not what they think it should show or might show if you recruit more cases.
There also does not seem to be a clearly defined set of hypotheses. I am hoping the research started with some, but it is really important to define the aims and hypotheses of the project in the introduction in order to define for the reader the scope of the work. There is some wording about the 'implications of stress on surgery', but your scope is in no way able to answer any question around this. What was your initial aim or aims here?
Some other more specific points:
The abstract needs to be more specific about what was found. For example, one of the stress scales was different between groups, but others were not. You also did not evaluate ‘pre-operative management’, so can’t say anything about it. Finally, what were the ‘main hypotheses’ that you refer to?
Line 47 – balanced anesthesia does not provide a ‘state of sedation’ because the patient is anesthetized.
Line 50 – you are correct that effective analgesia should be in place before the painful stimulus occurs, but what does that have to do with your study? Trazodone is not considered an analgesic.
Line 69 – you are certainly not medicating ‘pet owners’ here.
Line 78 – this is one reference that says you should not exceed 300 mg, and there was no evidence to back that statement up in your reference – it was just an opinion based on human medicine. Please remove or modify that statement.
Line 81 – trazodone should not be combined with many ‘behavior modifying drugs’ such as SSRIs due to the risk of serotonin syndrome. It is therefore not safe when combined with many of these drugs. Please modify this statement or remove.
Line 83 – your study is not exploring the implications of stress on anything, you are evaluating the administration of trazodone.
Line 85 – Implications are not hypotheses – please state your aims and hypotheses in a scientifically testable manner.
Line 96 – you state that all animals underwent a complete physical evaluation, but then state that 2 patients in the control group could not be examined due to aggression. These 2 patients need to be removed from the study and either replaced or the numbers that completed stated appropriately. You have no idea if these patients had a normal physical exam or not.
Line 133 – if the incision was caudal to the umbilical scar, it was not a ‘cranial incision’
Line 181 – your ‘adaptation’ of the stress scale appears to be merely a change in language. Did you translate the CDSS into another language and then translate it back to English? Because it is largely unreadable now and makes little sense in English. For example, you note ‘tense jaw’ under respirations, but then the same level in ‘mouth’ is tense – that is a tense jaw. “Don’t look at the VS but stalks the place’ is not in English at all, I don’t know what you are saying here. I am assuming you didn’t adapt the scale but rather used the scale in a different language?
Line 201 – You didn’t modify the REF, you merely used a single section of the scale – please note this appropriately in the manuscript.
Line 212 – according to my math, the maximum score is a 20 (one of the categories cannot score a 3, 7X3-1 = 20)
Line 232 – how did you administer the propofol – at a constant rate, in 0.5 mg/kg boluses, other? This will change the final dose.
Lines 245/246 – how did you measure MV and Raw? I don’t see tidal volume on here, so not sure how these were calculated? We have several of those monitors and you would need to manually enter the tidal volume in order to calculate MV, and they don’t do Raw at all.
Section 2.9 – you already noted that these parameters were recorded – please remove this section.
Statistical analysis – why didn’t you evaluate parameters over time? I would suggest linear mixed models would be better able to handle the overall analysis both within and between groups.
Line 280 – Why was this one animal excluded. And, if you excluded one and added another one, then you screened 17, not 16
Lines 304-305 – If you are unable to assess BP in the awake animals (and respiratory rate in most of them), please just remove this evaluation all together.
For the chem/CBC results, please just note that there were no differences between groups and move on. The tables do not provide any extra information and can be removed.
Line 326 – there is no such thing as a trend in statistical results – either the groups were different or they were not. You have a-priori set a value of 0.05 as defining whether these groups are different or not – therefore there was no difference in these groups. Same goes for many of the results moving forward. When you start saying that there were not significant differences, but you think that it looks like there should have been, it obviates the need for analysis. Please stick to the facts and your analysis – not what you think is a trend or ‘notable’
Line 349 – a mean score for what?
Line 367 – what is HT?
Line 374 – was this SAP increase statistically significant?
Line 377 – why were the ETCO2 values different between groups – you were controlling ventilation, so that is a ‘you’ problem rather than a ‘patient’ problem
Lines 394/395 – was this difference significant?
Line 545 – I don’t think you were powered to evaluate safety in this study?
Author Response
Response to Rewiever 1 comments:
We truly thank the reviewer for the support and all the comments and suggestions that help us to improve our manuscript. All detailed review reports were revised and changed. The changes were highlighted in the manuscript.
Point 1: The authors present an interesting report on the effects of trazodone administration prior to veterinary visits that involve general anesthesia. The patient numbers are quite low, and there does not appear to be a power analysis or justification for why 16 patients total were thought to be enough. There also seems to be quite a bit of interpretation outside of the actual statistical analysis - if 2 groups are not different based on your pre-determined analysis strategy, then they are not different... There is no such thing as a trend when rigorously evaluating 2 groups for differences - they are either different or they are not. The authors need to revise their language to reflect what the data shows, not what they think it should show or might show if you recruit more cases.
Response 1: We sincerely thank the reviewer for their insightful comments. We acknowledge that we did not perform a power analysis or sample size evaluation, and we recognize that the small sample size is a significant limitation of our study. For this reason, we explicitly stated in the title that this is a preliminary study. The primary aim of this work was to explore potential differences and generate hypotheses for future, larger-scale investigations. We fully agree on the importance of power calculations for confirmatory studies, and we will incorporate these in future research to ensure adequate sample sizes.
Regarding the interpretation of the results, we appreciate the reviewer's emphasis on rigor and objectivity. We agree that statistical analysis should strictly adhere to pre-determined methods, and we will revise our language to ensure it accurately reflects the data without overinterpretation. Specifically, we will remove any references to "trends" or speculative statements that extend beyond the statistical findings. Our goal is to present the results transparently and let the data speak for itself.
Once again, we thank the reviewer for their constructive feedback, which will undoubtedly improve the quality and clarity of our manuscript.
Point 2: There also does not seem to be a clearly defined set of hypotheses. I am hoping the research started with some, but it is really important to define the aims and hypotheses of the project in the introduction in order to define for the reader the scope of the work. There is some wording about the 'implications of stress on surgery', but your scope is in no way able to answer any question around this. What was your initial aim or aims here?
Response 2: We fully understand the importance of formulating a clear and well-defined hypothesis; therefore, we have modified the introduction in the manuscript. We hope that the revised text is now more suitable for publication.
Some other more specific points:
Point 3: The abstract needs to be more specific about what was found. For example, one of the stress scales was different between groups, but others were not. You also did not evaluate ‘pre-operative management’, so can’t say anything about it. Finally, what were the ‘main hypotheses’ that you refer to?
Response 3: Thank you so much for your aprettiation, now we understand that could be confusing as it was not correctly explained in the manuscript. It has been modified to specify the findings of each evaluated parameter and remove statements not supported by the obtained data.
It has been specified that only one of the stress scales showed significant differences between groups, eliminating ambiguous terms like "trend" and references to preoperative management, which was not evaluated.
Point 4: Line 47 – balanced anesthesia does not provide a ‘state of sedation’ because the patient is anesthetized.
Response 4: Thank you very much for your suggestion, we have changed it into the manuscript.
Point 5: Line 50 – you are correct that effective analgesia should be in place before the painful stimulus occurs, but what does that have to do with your study? Trazodone is not considered an analgesic.
Response 5: Thank you very much for your suggestion, the mention that preoperative analgesia is related to this study has been removed, as trazodone is not an analgesic.
Point 6: Line 69 – you are certainly not medicating ‘pet owners’ here.
Response 6: We have alredy corrected into the manuscript. We hope the text will be more suitable now. Sorry for the inconvenience.
Point 7: Line 78 – this is one reference that says you should not exceed 300 mg, and there was no evidence to back that statement up in your reference – it was just an opinion based on human medicine. Please remove or modify that statement.
Response 7: We are very grateful for your comment. The statement about trazodone's safety in combination with other drugs has been removed.
Point 8: Line 81 – trazodone should not be combined with many ‘behavior modifying drugs’ such as SSRIs due to the risk of serotonin syndrome. It is therefore not safe when combined with many of these drugs. Please modify this statement or remove.
Response 8: We are very grateful for your comment. The statement about trazodone's safety in combination with other drugs has been removed.
Point 9: Line 83 – your study is not exploring the implications of stress on anything, you are evaluating the administration of trazodone.
Response 9: Thank you for pointing this out. In this manner, the study hypothesis has been rewritten to focus exclusively on trazodone administration and its effect on the evaluated parameters.
Point 10: Line 85 – Implications are not hypotheses – please state your aims and hypotheses in a scientifically testable manner.
Response 10: We sincerely appreciate the reviewer’s insightful comment. We have revised the manuscript to ensure that the hypotheses are stated in a scientifically testable manner. We believe this modification enhances the clarity and scientific rigor of the manuscript. Thank you for your valuable feedback.
Point 11: Line 96 – you state that all animals underwent a complete physical evaluation, but then state that 2 patients in the control group could not be examined due to aggression. These 2 patients need to be removed from the study and either replaced or the numbers that completed stated appropriately. You have no idea if these patients had a normal physical exam or not.
Response 11: We completely understand your point. To clarify, one of the bitches was successfully examined; however, rectal temperature measurement and non-invasive blood pressure assessment had to be omitted. Nonetheless, she was still considered valid for inclusion in the study. The other bitch was indeed replaced to ensure the integrity of the study
We sincerely apologize for any confusion this may have caused. Thank you very much for your careful review.
Point 12: Line 133 – if the incision was caudal to the umbilical scar, it was not a ‘cranial incision’
Response 12: We completely agree, the description of the surgical incision has been corrected to accurately reflect its anatomical location and we hope it would be more suitable now.
Point 13: Line 181 – your ‘adaptation’ of the stress scale appears to be merely a change in language. Did you translate the CDSS into another language and then translate it back to English? Because it is largely unreadable now and makes little sense in English. For example, you note ‘tense jaw’ under respirations, but then the same level in ‘mouth’ is tense – that is a tense jaw. “Don’t look at the VS but stalks the place’ is not in English at all, I don’t know what you are saying here. I am assuming you didn’t adapt the scale but rather used the scale in a different language?
Response 13: We sincerely appreciate the reviewer’s careful attention to this matter and the valuable feedback provided. Upon reviewing the concerns raised, we have modified the table accordingly. Rather than an adaptation, the table is now a direct reproduction of the original Clinic Dog Stress Scale (CDSS) from King et al. (2022), ensuring that the terminology remains accurate and consistent with the source. This modification eliminates any unintended alterations in meaning or clarity. We are grateful for the reviewer’s insights, which have helped improve the accuracy and readability of our manuscript. Thank you again for your thoughtful comments.
Point 14: Line 201 – You didn’t modify the REF, you merely used a single section of the scale – please note this appropriately in the manuscript.
Response 14: The description of the REF scale has been corrected, specifying that only one section of it was used.
Point 15: Line 212 – according to my math, the maximum score is a 20 (one of the categories cannot score a 3, 7X3-1 = 20)
Response 15: We apologize for the inconvenience, the maximum score in the sedation scale has been corrected.
Point 16: Line 232 – how did you administer the propofol – at a constant rate, in 0.5 mg/kg boluses, other? This will change the final dose.
Response 16: Thank you for pointing this out. The propofol administration protocol has been specified to clarify the methodology used. In our study, propofol was administered in increments of 0.5 mg/kg IV boluses over 15 seconds until orotracheal intubation was possible, with a maximum total dose of 3 mg/kg IV. This protocol was similar to that described by Raszplewicz et al. We reviewed and included this information in the manuscript. We hope that it will be more suitable now.
Raszplewicz J, Macfarlane P, West E. Comparison of sedation scores and propofol induction doses in dogs after intramuscular premedication with butorphanol and either dexmedetomidine or medetomidine. Vet AnaesthAnalg. 2013 Nov;40(6):584-9. doi: 10.1111/vaa.12072. Epub 2013 Jul 25. PMID: 23889781.
Point 17: Lines 245/246 – how did you measure MV and Raw? I don’t see tidal volume on here, so not sure how these were calculated? We have several of those monitors and you would need to manually enter the tidal volume in order to calculate MV, and they don’t do Raw at all.
Response 17: We have reviewed and corrected this paragraph in the methodology section, as there was an error on our part. When drafting the methodology, we inadvertently referenced a previous study conducted by our research group three years ago, in which pulmonary compliance and airway resistance were evaluated after the establishment of pneumoperitoneum in laparoscopic ovariectomy. Indeed, these data were obtained using an advanced mechanical ventilation system, which in this case was likely the Mindray Anaesthesia Machine WATO EX-65 Pro.
In the present study, however, we used a MindrayAnaesthesia Machine WATO EX-35 Vet, and although various ventilatory parameters such as tidal volume, minute volume, peak pressure, plateau pressure, respiratory rate, and PEEP were recorded, these were finally not included in the statistical analysis. We sincerely apologize for any confusion this may have caused. Thank you very much for your careful review.
Point 18: Section 2.9 – you already noted that these parameters were recorded – please remove this section.
Response 18: We absolutely understand your point. The repeated mention of already noted parameters has been removed and we hope it would be more suitable now.
Point 19: Statistical analysis – why didn’t you evaluate parameters over time? I would suggest linear mixed models would be better able to handle the overall analysis both within and between groups.
Response 19: We thank the reviewer for their valuable comment. In our study, we opted to use traditional statistical methods, such as repeated measures ANOVA and the Friedman test, due to their simplicity and common use in similar previously published studies. Additionally, this approach was chosen in consultation with a professional statistician, who recommended these methods based on the study design and data characteristics. However, we acknowledge that linear mixed models could offer greater flexibility and robustness in the analysis. In future studies, we will consider implementing linear mixed models to enhance the statistical analysis. We really appreciate this suggestion and will take it into account for future research.
Point 20: Line 280 – Why was this one animal excluded. And, if you excluded one and added another one, then you screened 17, not 16
Response 20: We sincerely appreciate the reviewer’s careful attention to this detail and for pointing it out. The animal was excluded because a complete physical examination was not possible due to aggressive behavior. Since this animal was replaced with another, a total of 17 animals were screened, although only 16 were ultimately included in the study. We have now clarified this point in the manuscript to ensure accuracy. Thank you again for your valuable input.
Point 21: Lines 304-305 – If you are unable to assess BP in the awake animals (and respiratory rate in most of them), please just remove this evaluation all together.
For the chem/CBC results, please just note that there were no differences between groups and move on. The tables do not provide any extra information and can be removed.
Response 21: We are sorry about that. The evaluation of blood pressure and respiratory rate in awake animals has been removed, as they could not be correctly recorded. Unnecessary tables in hematological and biochemical analysis have been reduced. Thank you for your comment.
Point 22: Line 326 – there is no such thing as a trend in statistical results – either the groups were different or they were not. You have a-priori set a value of 0.05 as defining whether these groups are different or not – therefore there was no difference in these groups. Same goes for many of the results moving forward. When you start saying that there were not significant differences, but you think that it looks like there should have been, it obviates the need for analysis. Please stick to the facts and your analysis – not what you think is a trend or ‘notable’
Response 22: As we expressed before, we fully agree with your comment that statistical results should be interpreted strictly. We acknowledge that any language suggesting "trends" or subjective interpretations beyond the statistical findings is inappropriate and undermines the rigor of the analysis.
We have committed to revising the manuscript to ensure that all interpretations are strictly based on the data and the results of the statistical tests. We will carefully review the text, particularly in line 326 and similar sections, to remove any references to "trends," "notable differences," or speculative statements. We sincerely appreciate the correction, as it has helped us improve the clarity and accuracy of our manuscript. We hope that these revisions will fully address your concerns.
Point 23: Line 349 – a mean score for what?
Response 23: Specification of which parameter the "mean score" refers to has been added to the text. Thank you for your valuable feedback.
Point 24: Line 367 – what is HT?
Response 24: We sincerely apologize to the reviewer for the wording error in the text. We have carefully corrected the mistake to ensure clarity and accuracy in the manuscript.
Point 25: Line 374 – was this SAP increase statistically significant?
Response 25: We sincerely appreciate the reviewer’s attention to this detail. Upon reviewing the data, the increase in SAP at T2 in the control group (113.25 ± 16.55 mmHg) compared to the trazodone group (106.57 ± 10.15 mmHg) was not statistically significant, as no specific statistical comparison between these values reached a p-value < 0.05. However, we acknowledge the importance of clearly specifying this in the text to avoid any ambiguity. We have now revised the manuscript to explicitly state that while an increase was observed, it did not reach statistical significance. Thank you for your valuable feedback.
Point 26: Line 377 – why were the ETCO2 values different between groups – you were controlling ventilation, so that is a ‘you’ problem rather than a ‘patient’ problema
Response 26: We appreciate the reviewer's observation regarding the differences in ETCOâ‚‚ values between the groups. Although ventilation was volume-controlled, slight variations in ETCOâ‚‚ can occur. In this case, we believe that the statistically significant difference observed may reflect random variation or minor physiological disparities between groups rather than a systematic error in ventilation management. Importantly, both groups remained within the normocapnic range (ETCOâ‚‚ between 35–45 mmHg), indicating that ventilation was adequately managed in all animals.
Given that the observed differences, while statistically significant, were not clinically relevant (as both groups were within the physiological range), we initially decided to omit this information from the main manuscript to maintain focus on the primary findings of the study. However, we will include this data in the supplementary table for transparency.
We acknowledge that these differences could be further explored in future studies to better understand their clinical relevance.
Point 27: Lines 394/395 – was this difference significant?
Response 27: We appreciate the reviewer's observation regarding the differences in temperature. Given that the observed differences, while statistically significant, were not clinically relevant for the study, we initially decided to omit this information from the main manuscript to maintain focus on the primary findings of the study. However, we will include this data in the supplementary table for transparency.
We acknowledge that these differences could be further explored in future studies to better understand their clinical relevance.
Point 28: Line 545 – I don’t think you were powered to evaluate safety in this study?
Response 28: We apologize for the inconvenience, statements regarding trazodone's safety have been modified as this study's sample size does not allow for a conclusive evaluation of safety. Thank you for pointing this out.
Thank you for your consideration. We look forward to hearing from you.
Kind regards,
Cambeiro-Camarero, Nerea.

Reviewer 2 Report
Comments and Suggestions for Authors
Dear Authors,
your paper titled A preliminary study: Evaluation of oral trazodone as a strategy 2 to reduce anesthetic requirements in bitches undergoing ovariectomy has new and interesting findings ta could improve stress managing of the animals often difficult to manage when conducted in hospital and submitted surgery.
Overall, I found the article very interesting but in some points a bit too redundant and repetitive. I also think it needs a linguistic revision.
Please find my comments regarding to the manuscript below:
SIMPLE SUMMERY:
- Line 19: delete from additionally to group.
- Line 26: add oral administration
- Line 28: add intravenous fentanyl ….
- Line 30: which parameters?
- Line 32: delete from No significant to observed
- Line 35: delete does not cause intraop. hypotension
MM
- Line93: Specify the number of approval
- Line 96: transfer all the information from line 148 to 176 here in a more concise way
- Line 124: leave only the dosage of propofol
- Line 131: specify how the dose of sevo was adjusted until the minimum required dose was reached
- Line 133-143: summarize the procedure in 2 lines (the dogs were undergoing ovariectomy via cranial midline laparotomy)
- line 186: delete Dogs with high scores were considered highly stressed, while dogs with a score of 0 were considered calm and relaxed
- line 231: delete (Lipuro, BBraun)
- line 252: specify how frequently
- line 260: hemodynamic parameters (SAP, DAP, and MAP) were recorded…..
- line 262-263: delete the period
- line 284: develop a table for the data from the line 284 to line 292
- line 362: IV fentanyl
- line 337: delete (a mixture of dexmedetomidine and morphine)
- line 395: delete Dexdomitor, Ecuphar
DISCUSSION
- the discussions need to be rewritten, trying to make a linear discussion starting from the hypothesis and the confirmations obtained with the study
Comments on the Quality of English Language
The English should be improved.
Author Response
Response to Reviewer 2 comments:
We are grateful for all the comments and suggestions that helped us to improve our manuscript. All detailed review reports were revised and changed. The changes were highlighted in the manuscript.
Point 1: Line 19: delete from additionally to group.
Response 1: We have modified into the manuscript. Thank you for the suggestion.
Point 2: Line 26: add oral administration
Response 2: We have revised and changed into the manuscript. Thank you for this point.
Point 3: Line 28: add intravenous fentanyl ….
Response 3: We have modified into the manuscript. Thank you for the suggestion.
Point 4: Line 30: which parameters?
Response 4: The parameters concomitant evaluated were EtCO2, end-tidal sevoflurane concentration, temperature and SpO2. We appreciate your point and we have added to the manuscript.
Point 5: Line 32: delete from No significant to observed
Response 5: We have modified into the manuscript. Thank you for the suggestion.
Point 6: Line 35: delete does not cause intraop. hypotension
Response 6: We have modified into the manuscript. Thank you for the suggestion.
Point 7: Line93: Specify the number of approval
Response 7: We completely understand the importance of the reference number for the ethical approval, that is why in response to your point we have added to the line 93 in the manuscript.
Point 8: Line 96: transfer all the information from line 148 to 176 here in a more concise way
Response 8: We have modified into the manuscript. Thank you for the suggestion.
Point 9: Line 124: leave only the dosage of propofol
Response 9: We have revised and changed into the manuscript. Thank you for this point.
Point 10: Line 131: specify how the dose of sevo was adjusted until the minimum required dose was reached
Response 10: We apologize for the inconvenience, there was a mistake on the redaction of line 131. We must have written that the anesthesia was maintained with sevoflurane (vaporizer setting at 2%) in 100% Oâ‚‚, with the concentration adjusted to achieve the appropriate anesthetic depth, which was determined by clinical indicators, including mandibular muscle tone, ventromedial rotation of the eyeball, and the absence of the palpebral reflex.
Ribeiro, L.M.; Ferreira, D.A.; Brás, S.; Gonzalo-Orden, J.M.; Antunes, L.M. Correlation between Clinical Signs of Depth of Anaesthesia and Cerebral State Index Responses in Dogs with Different Target-Controlled Infusions of Propofol. Vet Anaesth Analg 2012, 39, 21–28, doi:10.1111/j.1467-2995.2011.00657.x.
We have modified on the manuscript. Once again we apologize for this mistake.
Point 11: Line 133-143: summarize the procedure in 2 lines (the dogs were undergoing ovariectomy via cranial midline laparotomy)
Response 11: We have summarized the procedure into the manuscript, we hope it would be more suitable now. Thank you for your point.
Point 12: line 186: delete Dogs with high scores were considered highly stressed, while dogs with a score of 0 were considered calm and relaxed
Response 12: We have modified into the manuscript. Thank you for the suggestion.
Point 13: line 231: delete (Lipuro, BBraun)
Response 13: We have modified into the manuscript. Thank you for the suggestion.
Point 14: line 252: specify how frequently
Response 14: As we did with other parameters we recorded the appearing of signs of intraoperative pain every five minutes. Moreover, anesthesia team was constantly monitoring the pacients in order to solve any painful episode administering a bolus of 2,5 µgr/kg IV fentanyl when necessary.
In order to clarify this point we have added the information to the manuscript. We are grateful for your sugestion.
Point 15: line 260: hemodynamic parameters (SAP, DAP, and MAP) were recorded…..
Response 15: We have modified into the manuscript. Thank you for the suggestion.
Point 16: line 262-263: delete the period
Response 16: We have modified into the manuscript. Thank you for the suggestion.
Point 17: line 284: develop a table for the data from the line 284 to line 292
Response 17: We have added the table into the manuscript. We are grateful for your sugestion.
Point 18: line 362: IV fentanyl
Response 18: We have modified into the manuscript. Thank you for the suggestion.
Point 19: line 337: delete (a mixture of dexmedetomidine and morphine)
Response 19: We have modified into the manuscript. Thank you for the suggestion.
Point 20: line 395: delete Dexdomitor, Ecuphar
Response 20: We have modified into the manuscript. Thank you for the suggestion.
Point 21: the discussions need to be rewritten, trying to make a linear discussion starting from the hypothesis and the confirmations obtained with the study
Response 21: We apologize for the inconvenience, we have rewritten some items of the Material and Method section and also on the Discussion in order to better adhere the text to the obtained results and to the main hypothesis, moreover we have hardly tried to write a more linear discussion. Thank you very much for your point.
We are grateful for all comments and suggestions that helped us to improve our manuscript. After some modifications according to the indications of the reviewers, we would like to re-submit our manuscript. We sincerely hope that the manuscript will be more suitable for publication.
Thank you for your consideration. We look forward to hearing from you.
Kind regards,
Cambeiro-Camarero, Nerea.

Reviewer 3 Report
Comments and Suggestions for Authors
This paper deals with an interesting topic for clinical practice and I appreciate that it must had been a huge work for the authors, congratulations! This is a wellwritten manuscript for a well designed work!
I just have a few considerations and doubts to ask the authors:
- The manuscript says that the tramadol is administered 2 hours before arriving at the hospital, however, it is easy to think that from the time the patient arrives at the hospital until the time the patient is premedicated, a very variable time can pass. Have they taken this variable into account to homogenize the study? In that case, I don't know if it could be clarified in a more precise way in the Material and Method section.
- On line 116, you must include the route of administration of the drugs. - It may also be advisable to clarify the surgeon's experience, since, although the technique is described in this way in the literature, it is common to tend towards supra-/infraumbilical minilaparotomies. This could also have repercussions on your study, since perhaps the traction of the ovary is slightly greater and the surgery time is shorter. - I also do not find it logical to take only one blood sample, since without prior intra-individual comparison it is difficult to compare between groups. Why was it decided to take only one sample, since this is a prospective study? - Another question I have is why did they wait so long from the administration of premedication until the induction of anesthesia? - Another question I have is why they waited so long from the administration of premedication until the induction of anesthesia? On the other hand, I also don't understand why they chose a dose of 5 microgr/kg of dexmedetomidine and not a lower one, in order to better assess the effects of the tramadone. - In the discussion they mention that one of the parameters considered to homogenize the study group was body condition. They should detail this better in the Material and Methods section, since it is not highlighted. - O line 425, I would remove the words "with dexmedetomidine and" since it is only morphine that will cause that adverse effect. - In paragraph lines 477 to 483, they mention that hypotension was not observed in their study, however, they compare it with another study in which they used acepromazine together with tramadone, which is more hypotensive than dexmedetomidine, and they also do not report the dose used in said study. I think they should rewrite this paragraph to clarify the differences between studies and whether something can be concluded with so much heterogeneity. -In the paragraph between lines 485 and 490, they conclude that since there was no bleeding in their study, the role of the tramadone in hemostasis can be ruled out. However, it must be taken into account that it is a simple surgical technique, in which there should not have been bleeding and, in addition, they have not even performed a hematology to assess possible changes. Nor have they mentioned the dose used in the other study with which they dispute. I do not think that one can be so forceful in such an assertion. - On the other hand, I don't know if it's a fault in my configuration but figures 2 and 3 do not appear in the manuscript. To sum up, I think this is a good study and it has clinical utility, but perhaps they have tried to cover too many parameters, which makes it very complicated to arrive at results to discuss. In addition, there are some shortcomings in the execution of the Materials and Methods section. The authors should approach the work by adhering more closely to the results obtained.
Author Response
Response to Reviewer 3 comments:
We truly thank the reviewer for the support and all the comments and suggestions that help us to improve our manuscript. All detailed review reports were revised and changed. The changes were highlighted in the manuscript.
Point 1: The manuscript says that the tramadol is administered 2 hours before arriving at the hospital, however, it is easy to think that from the time the patient arrives at the hospital until the time the patient is premedicated, a very variable time can pass. Have they taken this variable into account to homogenize the study? In that case, I don't know if it could be clarified in a more precise way in the Material and Method section.
Response 1: Thank you so much for your aprettiation, now we understand that could be confusing as it was not correctly explained in the manuscript.
To answer the question, we may say all patients were cited in different days at the same hour, when the surgery and anesthesia team was fully available for the study procedures, then the procedures started as soon as patients arrived at the hospital. That is why we do not considered that time lapse as a variable. However we think that you are completely right and we will take this variable into account for the next time.
In order to clarify this point we included an explanation on the Material and Method section.
Point 2: On line 116, you must include the route of administration of the drugs.
Response 2: Thank you very much for your suggestion, we have added it to the manuscript.
Point 3: It may also be advisable to clarify the surgeon's experience, since, although the technique is described in this way in the literature, it is common to tend towards supra-/infraumbilical minilaparotomies. This could also have repercussions on your study, since perhaps the traction of the ovary is slightly greater and the surgery time is shorter.
Response 3: We completely agree with your point. In first instance, to clarify the experience of surgeons, we should add that the surgery and anesthesia team was composed by highly qualified profesionals, with large experience and who are also part of the university proffesor staff.
In the other hand, similar to that published by Shariati et al. where a 4 to 6 cm skin incision was made in dogs with a mean weight of 14 kg, we made an incision of a length of approximately 4-8 cm, as some bitches of our study were truly large (31 kg) and with a deep chest conformation. Nevertheless, I would like to apologize and I must add that for next studies we are going to work with minimally invasive techniques as laparoscopic.
(Shariati, E.; Bakhtiari, J.; Khalaj, A.; Niasari-Naslaji, A. Comparison between Two Portal Laparoscopy and Open Surgery for Ovariectomy in Dogs. Vet. Res. forum an Int. Q. J.2014, 5, 219–223).
Point 4: I also do not find it logical to take only one blood sample, since without prior intra-individual comparison it is difficult to compare between groups. Why was it decided to take only one sample, since this is a prospective study?
Response 4: We appreciate your suggestion and we completely agree. In first instance we may say this was due to budget issues. By the moment the study is not and was not financiated so the animal´s guardians afford all costs of the surgery, drugs and analytical parameters. Based on previously studies of our own, we decided to save the money of a second blood sample analysis. In the other hand, we agree with you as we are adding a new drug in the procedure and this second blood analysis could helped us to clarify some aspects on the discussion. We will take this into account for next studies. Thank you very much.
Fernández-Martín, S.; Valiño-Cultelli, V.; González-Cantalapiedra, A. Laparoscopic versus Open Ovariectomy in Bitches: Changes in Cardiorespiratory Values, Blood Parameters, and Sevoflurane Requirements Associated with the Surgical Technique. Animals (Basel) 2022, 12, doi:10.3390/ani12111438.
Point 5: Another question I have is why did they wait so long from the administration of premedication until the induction of anesthesia? Another question I have is why they waited so long from the administration of premedication until the induction of anesthesia?
Response 5: We completely understand your point. We would like to explain that the hospital where the procedures were executed was an universitary hospital so it taked a little longer explaining the procedures to the students while we worked on them. Although, we will take this into account and we will try to work more efficiently for the next time.
Point 6: On the other hand, I also don't understand why they chose a dose of 5 microgr/kg of dexmedetomidine and not a lower one, in order to better assess the effects of the tramadone.
Response 6: We are very grateful for your comment. Presently we understand that 5 µgr/kg IM is a large dose and we will work with lower doses for next studies. As an explanation we may say that we were used to work with those doses for rutine at the hospital without have observed side effects and Grubb et al. described a dose range between 3 – 7 µgr/kg IM to achieve moderate sedation.
​​Grubb, T.; Sager, J.; Gaynor, J.S.; Montgomery, E.; Parker, J.A.; Shafford, H.; Tearney, C. Anesthesia and Monitoring Guidelines for Dogs and Cats. J Am Anim Hosp Assoc 2020, 56, 59–82.
Pan, S.Y.; Liu, G.; Lin, J.H.; Jin, Y.P. Efficacy and Safety of Dexmedetomidine Premedication in Balanced Anesthesia: A Systematic Review and Meta-Analysis in Dogs. Animals 2021, 11. ​
Point 7: In the discussion they mention that one of the parameters considered to homogenize the study group was body condition. They should detail this better in the Material and Methods section, since it is not highlighted.
Response 7: Thank you for pointing this out. In this manner, we have rewritten the manuscript adding that a scale of one to five was used, with one been thin and five been obese. Additionally, we have added more bibliography to clarify how we standardized the BCS. Thank you for the suggestion.
WSAVA Nutritional Assessment Guidelines. J Feline Med Surg 2011, 13, 516–525, doi:10.1016/j.jfms.2011.05.009.
Impellizeri, J.A.; Tetrick, M.A.; Muir, P. Effect of Weight Reduction on Clinical Signs of Lameness in Dogs with Hip Osteo-arthritis. J Am Vet Med Assoc 2000, 216, 1089–1091, doi:10.2460/javma.2000.216.1089.
Point 8: O line 425, I would remove the words "with dexmedetomidine and" since it is only morphine that will cause that adverse effect.
Response 8: We completely agree. We have romeved those words from the manuscript and we hope it would be more suitable now.
Point 9: In paragraph lines 477 to 483, they mention that hypotension was not observed in their study, however, they compare it with another study in which they used acepromazine together with tramadone, which is more hypotensive than dexmedetomidine, and they also do not report the dose used in said study. I think they should rewrite this paragraph to clarify the differences between studies and whether something can be concluded with so much heterogeneity.
Response 9: We absolutely understand your point. Since we reviewed the paragraph we notice a wrong explanation of the terms. In order to correct that, we have rewritten the paragraph in the manuscript according to the results and conclusions achieved on the study of Murphy et al. Thank you for your clarification.
Murphy, L.A.; Barletta, M.; Graham, L.F.; Reichl, L.J.; Duxbury, M.M.; Quandt, J.E. Effects of Acepromazine and Trazodone on Anesthetic Induction Dose of Propofol and Cardiovascular Variables in Dogs Undergoing General Anesthesia for Orthopedic Surgery. J Am Vet Med Assoc 2017, 250, 408–416, doi:10.2460/javma.250.4.408.
Point 10: In the paragraph between lines 485 and 490, they conclude that since there was no bleeding in their study, the role of the tramadone in hemostasis can be ruled out. However, it must be taken into account that it is a simple surgical technique, in which there should not have been bleeding and, in addition, they have not even performed a hematology to assess possible changes. Nor have they mentioned the dose used in the other study with which they dispute. I do not think that one can be so forceful in such an assertion.
Response 10: We apologize for the inconvenience, we fully agree with you, it is not correct to make an assertion about hemostasis as we did, without cuantitatively evaluate those parameters. About the item of have not performed second blood analysis we again apologize. At that time, we correct the paragraph on the manuscript, adding the doses used on the study and avoiding been determinant. We hope it would be more suitable now.
Point 11: On the other hand, I don't know if it's a fault in my configuration but figures 2 and 3 do not appear in the manuscript.
Response 11: We are sorry about that. We will make sure that all figures are correctly added to the manuscript from now on. Thank you for your comment.
Point 12: To sum up, I think this is a good study and it has clinical utility, but perhaps they have tried to cover too many parameters, which makes it very complicated to arrive at results to discuss. In addition, there are some shortcomings in the execution of the Materials and Methods section. The authors should approach the work by adhering more closely to the results obtained.
Response 12: Once again sorry for the inconvenience, we have rewritten some items of the Material and Method section and also on the Discussion in order to better adhere the text to the obtained results. Thank you very much for your point.
We hope the manuscript will be more suitable now.
Kind regards
Cambeiro-Camarero, Nerea

Round 2
Reviewer 2 Report
Comments and Suggestions for Authors
Dear Authors,
I am satisfied for the revisions made and for the improvements made to the manuscript.
Best regards.
Reviewer 3 Report
Comments and Suggestions for Authors
First of all, thank you for your kind response and after having made the relevant corrections, thus improving the manuscript, I give the go-ahead for publication.
Congratulations to the authors!